# Landscape Changes and Optimization in an Ecological Red Line Area: A Case Study in the Upper Reaches of the Ganjiang River

**DOI:** 10.3390/ijerph191811530

**Published:** 2022-09-13

**Authors:** Guangxu Liu, Aicun Xiang, Yimin Huang, Wen Zha, Yaofang Chen, Benjin Mao

**Affiliations:** School of Geography and Environmental Engineering, Gannan Normal University, Ganzhou 341000, China

**Keywords:** landscape pattern, ecological red line area, patch, community succession, the Ganjiang River

## Abstract

The key to optimizing ecological management is to study the spatial configuration of the landscape and the dynamic changes and their driving mechanisms at the landscape scale. The ecological red line area in the hilly area of the upper reaches of the Ganjiang River was chosen as the research area in this study. Based on the theory of landscape ecology and the evolution of biological communities, a multiscale coupling model was adopted and combined with remote sensing (RS) and geographical information system (GIS) technologies to systematically study the evolution of key landscape ecosystems such as forests, patch characteristics, and changes in diversity. The study revealed that: (1) forests represented the largest proportion in the study area, followed by croplands and grasslands; (2) the biological community tended to progress toward climax between 1986 and 1995, but then it moved toward regressive successions between 1995 and 2005 before recovering; (3) the study area was characterized by a high proportion of dominant ecosystems, most of which were at their climax with stable ecological species groups, and which were connected by ecological corridors; and (4) during the period from 1995 to 2010, most landscapes showed a trend of fragmentation. However, during the period from 2010 to 2018, the forest patches were gradually connected. The proportion of dominant landscapes increased, and the landscape uniformity was reduced. Based on the findings, we proposed an ecosystem management strategy that includes strengthening crop management, focusing on the natural restoration of the ecosystems and the cultivation of large patches, exploring disturbances due to mining activities, and applying methods to mitigate damage to and optimize the ecosystem.

## 1. Introduction

As mosaics of ecosystems [1], landscapes are extensive geographical features with distinct visual characteristics [2]. Driven by the problems posed by landscape changes, landscape ecology has become the foundation of landscape management, design, and policy making. Landscape changes such as spatial configuration are key to conserving biodiversity and ecosystem services [3,4]. Research shows that the effectiveness of fragmented protected areas is influenced by how easily wildlife can move within the landscape [5]. A new transition of landscape management and design is toward multifunctional landscapes [6], which drives research on landscape-related issues that continue to emerge and includes examinations of structures, functions, changes, scales, resilience [7], sustainability [8], approaches [9], designs [10], diagnosing the transformation [11], and regional landscape planning [12]. The landscape pattern is the spatial layout and the combination of landscape elements of various sizes and shapes [13], including the types, quantity composition, space configuration, diversity, fragmentation, connectivity, and dominance of patches, corridors, ecological matrixes, and elements [14]. The combined effects of natural and human factors can lead to changes in landscape patterns. For example, the impacts of human activities on landscape heterogeneity is manifested as the fragmentation and dispersion of landscape elements, biodiversity loss, and making landscape types more consistent and homogeneous [15]. Mountain forests planted in the 20th century are more susceptible to disturbance by natural factors such as wind resistance, bark beetle outbreaks, and forest fires than forests that existed before 1880 [16]. The study of dynamic changes in landscape spatial patterns and their driving forces has attracted more attention in landscape ecology research [17], with the objective of revealing the causes and processes of their evolution using regression analyses, multivariate statistical analyses, spatial autocorrelation analyses, artificial neural network analyses, and modeling methods in order to predict future pattern changes [18]. For example, Chen used a transition matrix and a landscape index to analyze the dynamic change in a forestscape pattern based on the interpretation of Landsat remote sensing images of three periods in 2009–2017, covering the study area of 2906 km^2^. She constructed an multicriteria evaluation–cellular automata–Markov chain (MCE-CA-Markov) model to simulate and predict the forestscape pattern of Ningxiang City in 2021 [19]. Chen et al., selected remote sensing images and used Fragstats 4.2 software to quantitatively analyze the landscape pattern, and they combined principal component analysis and a landscape comprehensive evaluation index to evaluate landscape pattern changes. They revealed the change process of the landscape pattern in the Changbai Mountains during the past 30 years [20]. To study the landscape pattern evolution in the Tiaozi mud reclamation area in Jiangsu, Yu et al., used remote sensing images and environmental and ecological survey data from 2010 to 2020 combined with landscape index changes and an area transfer matrix model to analyze the driving factors of landscape pattern evolution [21].

The exploration of “ecological red lines (ERLs)” began in 2005 in China [22]. ERLs are not geometric lines in this paper, but refer to strictly protected areas with minimum space and maximum human activity limits designated by the governments, which need special protection to maintain ecosystem diversity and connectivity to support national or regional ecological safety and sustainable development [23]. ERLs were included as a normative concept in the *Decision of the Central Committee of the Communist Party of China on Some Major Issues Concerning Comprehensively Deepening the Reform*, which was published in 2013 [24]. In May 2017, the Ministry of Environmental Protection and the National Development and Reform Commission jointly issued the *Guidelines for the delimitations of Ecological Protection Red Lines* [25]. Based on these guidelines, Chinese researchers have used GIS and other methods to delimit ecological red lines in different regions based on ecosystem theory (Table 1). In February 2017, following the release of *Several Opinions on Delimiting and Strictly Observing the Red Lines of Ecological Protection*, which was jointly issued by the Central Committee of the Communist Party of China and the State Council [26], China preliminarily completed the delimitations of ERLs in the whole country in 2021 [27]. Research and discussions regarding ERL protection have been promoted to a “national work” [28]. To meet the demands of the scientific and rational management of ERLs, research on scales such as urban areas [29], counties [30], and even small watersheds [31] has become the basis and key of ecological protection and control systems in ERLs [32].

On 30 June 2018, People’s Government of Jiangxi Province issued the *Plan for the delimitations of Ecological Protection Red Lines in Jiangxi Province*, which reported that the ERL in Jiangxi Province was 46,876.00 km^2^, representing 28.06% of the province’s land area [38]. Based on current research and progress, the ERL in the upper reaches of the Ganjiang River (UR-GJ) in Jiangxi Province was selected as the case area in this study. This study focuses on the dynamic changes and driving mechanisms of landscape patterns and functions. The study reveals the pattern characteristics, changes, and evolution of the primary landscapes in the study area using data analysis, sensitivity analysis, and other analysis methods. The results of the study provide scientific evidence for the region to formulate a multiobjective management model for the ERLs, optimize the ecosystem structure, and improve its ecosystem service functions. It also provides new information regarding ecosystem theory and management practice.

## 2. Data and Methods

### 2.1. Study Area

The UR-GJ are located in the south of Jiangxi Province between N24°29′–N27°09′ and E113°54′–E116°38′. The study area is the key to the ERL of Jiangxi Province, covering an area of 14,875 km^2^ and accounting for 31.7% of the total ERL. It includes the water conservation ecological protection red line and the Nanling mountainous biodiversity maintenance red line (Figure 1). The terrain is primarily mountainous and hilly. The rivers run from the south, east, and west, converging on the Ganzhou Basin, and flowing into the Ganjiang River in the north. The major rivers include the west source, the Zhangshui water system (the Zhangshui and Shangyou River), and the east source, the Gongshui water system (including Xiangshui, Lian River, Meijiang, Pingjiang, and Taojiang). The study area is significantly influenced by the subtropical humid monsoon climate system, with four distinct seasons that include one with abundant rainfall (March–June), a very hot summer (July–August–September), and an overall mild climate (Figure 1). The average annual rainfall is greater than 1300 mm and concentrated from March to June, which often causes large-scale floods [39].

The study area supports rich biodiversity, drawing from a long ecological history that has given Jiangxi Province a special status in China, with three national nature reserves, one provincial nature reserve, and 25 municipal or county-level nature reserves (Figure 1). The forests, whose coverage rate reaches as high as 82% in the ERLs, are primarily secondary forests and artificial forests, with obvious vertical distributions of tree species and complex forest structures. The study area contributes to three significant ecosystem services that provide important protection though conserving biodiversity, watersheds, and soils. It also provides ecosystem services of disaster resistance, dust removal, carbon fixation, and oxygen production. It is an extremely important area for maintaining the ecological security of the Ganjiang River and even that of Jiangxi Province.

### 2.2. Data

Landsat remote sensing images with resolutions of 30 m from 1986, 1995, 2005, 2010, and 2018 were selected for this study. They were interpreted, classified, and verified into six landscape types: croplands, forests, grasslands, water, urban or rural settlements, and others, with an overall classification accuracy of greater than 85% (Figure 2). Among the six landscape types, forests, grasslands, and water are closely related to soil conservation, water conservation, and biodiversity in the ERL zone, and settlements and croplands represent human disturbance to the natural ecosystems. According to the area, proportion (in Table 2), and spatial distribution of various landscapes in the study area, it can be seen that the area of forests, croplands, and grasslands accounts for the majority of landscape types. In 2018, the total area of forests was 12,201 km^2^, accounting for 82% of the total. The total crop area was 1677.2 km^2^, accounting for 11.3% of the total. The total grassland area was 762 km^2^, accounting for 5.1% of the total. The total water area was 180.5 km^2^, accounting for 1.2% of the total. The settlement area was 64.5 km^2^, accounting for 0.4% of the total.

As can be seen in Table 2 and Figure 2, during the 33 years from 1986 to 2018, the forest area remained above 12,200 km^2^, and was the primary ecological landscape. The area of grasslands and settlements somewhat decreased during this period. The area of water increased, with slight fluctuations. Other landscapes accounted for a very small proportion of the area, with no significant change. In general, the ecological environment is good in the study area, which provides strong support for its stability as a carbon sink, for biodiversity conservation, and for water conservation. However, the proportion of croplands is relatively high and increasing, indicating that human activities still have strong disturbances to the ecosystem.

### 2.3. Methods

As ecological units, Landscapes are heterogeneous land areas consisting of three fundamental elements: patches, corridors, and a matrix [40], of which patches are areas of connected homogeneous land-cover type that differs from its surroundings [41]. They can be examined for spatial patterns and statistical measures, which are telling of landscape structure. This paper investigated the changes of the landscapes with the Sankey map using GIS. The landscape patterns were analyzed with ecological indices calculated with Fragstats software at the patch scales and the landscape scales separately. The framework is shown in Figure 3. Below are the indices and the methods to calculate them.

#### 2.3.1. Ecological Indices at the Patch Scales

Eight types of ecological indices at the patch scales, namely the total class area, number of patches, percentage of patches, patch density, largest patch index, perimeter–area fractal dimension index, aggregation index, and patch cohesion index were selected and calculated as follows:(1)Total class area (*CA*)
(1)$$CA=\sum_{j=1}^{n}a_{ij}\frac{1}{10000}$$
where *a_ij_* is the area of patch *i* in landscape *j*. *CA* measures the composition of landscape. It equals the sum of the areas of all patches of the corresponding patch type, namely total class area. *CA* approaches 0 when the *i*th patch becomes increasingly rare in the *j*th landscape. *CA* equals the total area when the entire image is comprised of a single patch. The size of the *CA* limits the number and abundance of species that live in such class.(2)Number of patches (*NP*), *NP* > 1
(2)$$NP=\sum n_{i}$$
where *n_i_* is the number of the *i*th patch. *NP* equals the total number of patches of a certain class. It reflects the spatial pattern of the landscape and is often used to indicate the heterogeneity of the entire landscapes. Its value has a good positive correlation with the fragmentation degree of the landscape. *NP* affects the spatial distribution and stability of species.(3)Percentage of patches (*PLAND*)(3)$$PLAND=100A^{-1}\sum_{i=1}^{n}a_{ij}$$
where *a_ij_* is the area of patch *i* in landscape *j*; *A* is the total landscape area. When its value approaches 0, the patch class becomes very rare in the landscape. When the value is equal to 100, it indicates that the whole landscape consists of only one class. *PLAND* measures landscape components and the relative area of a patch class in the landscape, influencing ecosystem indicators such as biodiversity, dominant species, and abundance.(4)Patch density (*PD*)(4)$$PD=\frac{n_{i}}{A}\times 10000\times 100,PD>0$$
where *n_i_* is the total number of patch *i* in the landscape; *A* is the total area of the landscape. *PD* reflects the number of patches per unit area and can compare the types of patch composition among landscapes of varying size.(5)Largest patch index (*LPI*)(5)$$LPI=\frac{\max(a_{ij})}{A}\times 100$$
where *a_ij_* is the area of patch *i* in landscape *j*. *LPI* is helpful to determine the spatial pattern or dominant type of the landscape.(6)Perimeter–area fractal dimension index (*PAFRAC*)(6)$$PAFRAC=\frac{\frac{2}{\left[n_{i}\sum_{j=1}^{n}\left(\ln p_{i}{}_{j}\ln a_{ij}\right)\right]-\left[\left(\sum_{j=1}^{n}\ln p_{i}{}_{j}\right)\left(\sum_{j=1}^{n}\ln a_{ij}\right)\right]}}{\left(n_{i}\sum_{j=1}^{n}\ln p^{2}{{}_{i}}_{j}\right)-{\left(\sum_{j=1}^{n}\ln p_{i}{}_{j}\right)}^{2}}$$where *a_ij_* is the area of patch *i* in landscape *j*; *p_ij_* is the perimeter of patch *i*; *n_i_* is the total number of patches in the landscape. *PAFRAC* is used to measure the shapes of patches or landscapes. When *PAFRAC* tends to 1, the shape of patches tends to square. As it approaches 2, the patch shape tends to be highly convoluted, with plane-filling perimeters.(7)Aggregation index (*AI*)(7)$$AI=\left[\frac{g_{ii}}{\max(g_{ii})}\right]$$
where *g_ii_* is the aggregation index of like adjacent patch numbers of type *i* based on single-count algorithm; max(*g_ij_*) is the maximum of *g_ii_*; *AI* reflects the nonrandomness or aggregation degree of various patch classes in the landscape, and can reflect the spatial configuration characteristics of landscape composition. If the landscape is composed of many discrete patches, *AI* is small, but when the landscape is dominated by a few large patches or the same type of patches are highly connected, *AI* is large.(8)Patch cohesion index (*COHESION*)(8)$$COHESION=\left[1-\frac{\sum_{i=1}^{m}\sum_{j=1}^{n}p_{ij}}{\sum_{i=1}^{m}\sum_{j=1}^{n}p_{ij}\sqrt{a_{ij}}}\right]{\left[1-\frac{1}{\sqrt{Z}}\right]}^{-1}\times 100$$
where *p_ij_* is the perimeter of patch *ij* measured by pixel surface; *a_ij_* is the area of the patch *ij* measured by pixels; Z is the total number of pixels in the landscape. *COHESION* measures the physical connectedness of the corresponding patch, and it is sensitive to the aggregation of the focal class below the percolation threshold.

#### 2.3.2. Ecological Indices at the Landscape Scales

In addition to the above indices, the indices at the landscape scale include the Shannon diversity index (*SHDI*) and the Shannon evenness index (*SHED*). The *SHDI* and *SHED* are calculated as follows:(9)Shannon diversity index (*SHDI*)(9)$$SHDI=-\sum_{i=1}^{m}p_{i}\times \ln p_{i}$$
where *p_i_* is the proportion of the landscape occupied by patch *i*. *SHDI* = 0 indicates that the whole landscape consists of only one patch (no diversity). The increase in *SHDI* indicates the increase in patch classes or the trend distribution of each patch class in the landscape. *SHDI* is a measurement based on information theory, which can reflect landscape heterogeneity and is especially sensitive to the unbalanced distribution of various patch classes in the landscape. In a landscape system, the richer the landscape is, the higher the degree of fragmentation is, the greater the uncertainty information content is, and the higher the SHDI value is.(10)Shannon’s evenness index (*SHEI*)(10)$$SHEI=\frac{-\sum_{i}^{m}\left(p_{i}+\ln p_{i}\right)}{\ln m}$$
where *p_i_* is the proportion of the landscape occupied by patch *i*; *m* is the number of patch types (classes) present in the landscape. *SHEI* = 0 indicates that the landscape consists of only one patch without diversity. *SHEI* = 1 indicates that all patch classes were evenly distributed and had the greatest diversity. *SHEI* is a powerful means of comparing diversity changes between the various landscapes or within the same landscape over time. When *SHEI* approaches 1, the dominance degree is low, indicating that there is no obvious dominant type in the landscape and all patch classes are evenly distributed in the landscape.

## 3. Results

### 3.1. Landscape Changes

Changes in the landscape types were analyzed using a bar graph, as shown in Figure 4. The forests, croplands, and grasslands are the three ecological landscapes with a high proportion of the ERLs. Among them, forests accounted for the highest proportion. The lowest proportion of forests was found in 1986 (81.6%), and the highest was in 1995 (82.4%). There was little change from 2005 to 2018. In 2010, the proportion of forests was slightly higher than that in 2005 (0.01%), and in 2018, it decreased slightly, by 0.2%. The total proportion of forests and grasslands increased from 86.72% in 1986 to 87.44% in 1995 and decreased slightly from 2005 to 2018. The proportion of croplands was approximately 11%, and the lowest proportion was found in 1986, when croplands accounted for 10.9% of the total area. The highest proportion was found in 2010, when the proportion of croplands was 11.4%. Water areas, settlements, and others accounted for approximately 2% of the total area. The proportion of water areas remained stable, at approximately 1.2%, and the proportion of settlements declined significantly in 1995 and increased slightly in 2010. Figure 4 verifies the above conclusion that the ecology of the study area is good, but there is obvious disturbance by human beings. After the ERL is demarcated, measures should be taken to reduce the impact of agricultural activities on the ecosystem by improving crop management.

### 3.2. Ecological Succession and Human Disturbance

Ecological succession is the process in which one ecosystem is replaced by another ecosystem in a landscape over time [42,43]. The succession analysis of the study area was helpful in understanding the changes in the ecological functions and the reasons behind these changes. According to Section 2.3, GIS software was used to calculate the ecological transfer matrix of the four periods: 1986–1995, 1995–2005, 2005–2010, and 2010–2018. After removing the results of no succession, a Python program was written to make a Sankey map to intuitively analyze the direction of the governing succession in the ERL (Figure 5).

Figure 5 shows the area and specific direction of the six landscapes from 1986 to 2018. According to Figure 5, there were various degrees of succession in the landscapes, among which forests were the largest, followed by croplands, grasslands, and water. From 1986 to 1995, 1147 hectares of forest retrograded to grasslands, 978 hectares were disturbed and changed to croplands, and 18 hectares changed to settlements. A total of 2985 hectares of grassland progressed and succeeded to forests, and 8 hectares became croplands. There were 1338 hectares of croplands that developed into forests, 519 hectares to grassland, 209 hectares into water, and 51 hectares changed into settlements. A total of 229 hectares of water developed into forests, 747 hectares into croplands, and 148 hectares into grasslands. According to the data, from 1986 to 1995, the succession of the biological community showed a trend toward the climax, and the function of the ecosystem was supplemented, improved, and developed. From 1995 to 2005, 3773 hectares of forests changed to croplands, 811 hectares to grasslands, 304 hectares to water, and 55 hectares to settlements. A total of 1766 hectares of grasslands developed to forests, 1607 hectares of grasslands became croplands, and 180 hectares became water. There were 1214 hectares of croplands that changed to water, 692 hectares to forests, 77 hectares to grasslands, and 21 hectares to settlements. There were 1214 hectares of croplands, 304 hectares of forests, 180 hectares of grasslands, and 2 hectares of settlements that changed to water. In general, a large number of forests were replaced by croplands during the period 1995–2005. Due to the construction of reservoirs and other projects, there was a clear succession of ecosystems into water. The ecological systems were notably disturbed by human activities during this period. From 2005 to 2010, 878 hectares were retrograded into grasslands, 2391 hectares of forests changed into croplands, 66 hectares into water, and 19 hectares into settlements. A total of 2079 hectares of grasslands changed into forests, 153 hectares into croplands, and 11 hectares into water. There were 1681 hectares of croplands that changed to forests, 126 hectares to grasslands, 116 hectares to water, and 61 hectares to settlements. A total of 61 hectares of settlements were replaced by forests, 65 hectares by croplands, and 10 hectares by grasslands. Water had 167 hectares replaced by croplands, 91 hectares by forests, and 15 hectares by grasslands. According to the data, from 2005 to 2010, there was a clear succession process to forests and grasslands. The ecological system was optimized, and the ecological functions obtained some degree of recovery. From 2010 to 2018, 40,954 hectares of forests were replaced by croplands, 14,539 hectares by grasslands, 1538 hectares by human settlements, and 2677 hectares by water. A total of 10,045 hectares of grasslands developed to forests, 4177 hectares to croplands, 384 hectares to settlements, and 431 hectares to water. A total of 41,181 hectares of croplands were replaced by forests, 4158 hectares by grasslands, 1701 hectares by human settlements, and 1324 hectares by water. In the human settlement area, 1365 hectares were replaced by croplands, 405 hectares by forests, and 93 hectares by grasslands. For the water, 2604 hectares were replaced by forests, 356 hectares by grasslands, 1546 hectares by croplands, and 76 hectares by settlements. During the period from 2010 to 2018, there was a phenomenon of small-scale expansion of human settlements to forests, grasslands, and other ecological areas, but the ecosystem in the study area remained stable overall.

The ecological succession during the various periods demonstrated that the natural ecosystem in the study area was less disturbed by human activities, and the proportion of landscape types in succession was small. These results indicated that the ecosystems were good and relatively stable in the ERLs. After being disturbed from 1995 to 2005, the ecosystems quickly recovered in approximately 2010. However, it should be pointed out that human settlements expanded to ecological areas from 2010 to 2018, which showed the obvious disturbance of human activities to the ecosystem. This may be due to rapid population growth in Ganzhou city during this period. Moreover, the Sankey map also showed that crop cultivation was an important perturbation factor that disturbed the ecosystem in the study area. From 1995 to 2005, a variety of landscapes were developed into croplands, and this might be an important reason for the interruption of ecosystem progress and succession. From 2005 to 2010, grasslands were the important contributor to ecological restoration. Under the human settlement expansion from 2010 to 2018, the transfer of large areas of croplands to forests was very important to maintaining the stability of the ecosystem in the study area.

### 3.3. Characteristics of the Landscape Patterns

The characteristics of landscape patterns and their impact on soil and water conservation and biodiversity in the ERL were analyzed using the landscape analysis software Fragstats 4.2. The results of the landscape indexes in 2018 are shown in Table 3. In 2018, forests covered a large area with 1853 patches, and this was the dominant patch class in the study area. The number of crop patches was 9420, the number of grassland patches was 2325, and the number of water patches was 423. The number of settlement patches was 1260. Patches of other landscapes accounted for the smallest area, and the number of patches was only four. The patch density (PD) of the croplands was 0.6, indicating that the spatial distribution of croplands was fragmented, and this was consistent with the mountainous and hilly topography and geomorphological characteristics of the study area. The ERL is not suitable for agricultural production activities such as crop cultivation. The PD of the forest patch was 0.1, indicating a large forest distribution and good ecological environment. The study area was suitable for maintaining biological diversity protection and water conservation. The LPI of forests was 7.9, while the LPIs of the other landscape classes were less than one, which further indicated that forests were the dominant landscape class. The PAFRAC values of the six landscapes were all greater than one. Among them, the PAFRAC values of croplands and water were both greater than 1.5, which demonstrated that the patch shapes were complex. The study area had more natural landforms, and had no regular morphological characteristics. In particular, the shape of the crop land was often affected by human activities. However, its morphological characteristics were still controlled by the natural topography and landforms in the ERL. This confirmed that the ERL was not suitable for large-scale mechanized farming. The largest COHESION index was that of forests. The COHESION indices of the grasslands and water were also greater than 90. These three were the primary habitats of organisms and the primary bearers of ecological function. The good spatial connectivity of the three patches showed that their ecological corridors were basically connected and conducive to animal activities between patches. The AI of forests was 92.5, and the AIs of the grasslands and forests were greater than 70, indicating that the three patches were primarily large patches, and that the same classes were connected. This ensured the stability and function of the ecosystem in the study area. The AI values of croplands and settlements were lower than those of other landscapes, indicating that human activities were more scattered, less connected, and less concentrated.

According to the analysis above, forests were the dominant landscape, and they had small PD, more contiguous distribution, better connectivity, and high aggregation sites. Grasslands and water were next, and they were dispersed through space, but they also had high connectivity and aggregation places. This indicated that the study area is the best habitat area in the hilly region of the UR-GJ, with large patch areas and a high proportion of dominant ecological types, most of which are succession climaxes. The ecological corridor is connected, the ecological components are stable, and the ecological function is good. It is an ideal biodiversity protection area and water conservation area. Croplands and settlements are spread naturally along the terrain, with fragmented patches and low connectivity between each other, which makes it difficult to form the aggregation effect and is not suitable for large-scale human production activities. After the ERL is demarcated, it will not have much impact on regional economic development.

### 3.4. Changes of the Ecological Indices at the Patch Scale

#### 3.4.1. Changes in the PD

Figure 6 shows the PD of six landscapes from 1986 to 2018. It can be seen from Figure 6 that the PD was the highest in croplands, followed by forests, grasslands, water, and settlements. Among the six landscapes, the PD of croplands was the highest, and it decreased from 0.66 in 1986 to 0.35 in 2010 and then increased to 0.63 in 2018. The PD of forests also had a large fluctuation; it was 0.13 in 1986, then increased to 0.33, and dropped to 0.12 in 2018. The PD of grasslands was the lowest; it was 0.03 in 1995, and the highest PD of grasslands was 0.36 in 2018 and 1986. The PD of the water was the highest in 1995, and all were below 0.05 in the other years. The PD of grasslands and settlements increased from 1995 to 2018. The PD of grasslands increased from 0.03 in 1995 to 0.16 in 2018. The PD reflects the number of patches per unit area, and an increase in the value indicates that patches gradually become smaller and shows a trend of fragmentation. According to the changes over the period, the PD of forests increased from 1995 to 2010, indicating an obvious trend of fragmentation. After China began paying attention to the construction of an ecological civilization, it gradually developed to being connected after 2010. After experiencing a decreasing trend, the grassland PD increased sharply in 2018 and showed an obvious trend of fragmentation in the later period. The PD values of the water areas were relatively stable. The big changes and high values of the cropland PD indicated that the intensity of human activities was driven by policy and the economy, but the overall cropland patch was relatively fragmented, which had a clear restrictive effect on agricultural development.

#### 3.4.2. Changes in the PAFRAC

The changes in the PAFRAC are shown in Figure 7. The largest PAFRAC value in 5 years was that of settlements, followed by that of croplands, and this represented the disturbance of human activities to the ecosystem. The high PAFRAC indicated high landscape complexity and a scattered distribution. The PAFRACs of grasslands, forests, and water were all below 1.5, indicating that the three landscapes were orderly and were the dominant landscape types in the study area. According to the trend, the PAFRACs of all the landscapes decreased from 2010 to 2018, which may have been the direct result of the government’s emphasis on ecological construction and the efforts to promote ecological civilization. From 1986 to 2010, the PAFRACs of the forests, grasslands, and water increased slightly, while those of croplands and settlements showed downward trends, which confirmed each other. This result indicates that the disturbance of human activities to the ecosystem increased and the invasion of the natural ecosystem by croplands and settlements increased.

#### 3.4.3. Changes in the AI

The changes in the AI from 1986 to 2018 are shown in Figure 8. In Figure 8, the AI of forests was the highest, and it was approximately 92 in the past 32 years. It decreased slightly from 1995 to 2010 and increased slightly to 92.5 from 2010 to 2018. The average AI of grasslands was approximately 73. The lowest was in 2010 and the highest was in 2018. The AI of water showed similar changes to that of the grasslands. The changes in the AI of the croplands and the settlements were complex. The AI of the croplands increased to 62.84 in 2005, decreased to 62.25 in 2010, and increased to 63.51 in 2018. The changes in the settlement AI were relatively stable, fluctuating near 70. Figure 8 shows that the dominant landscape of forests has been relatively stable, and their niche has been enhanced. The concentration of settlements and croplands has been increasing since 2010, and the connectivity between patches has also improved.

### 3.5. Changes of the Ecological Indices at the Landscape Scale

Table 4 shows the ecological indices at landscape scales from 1986 to 2018. The total number of patches (NP) in the study area remained stable from 1986 to 2018, fluctuating between 15,000 and 15,800. The lowest NP was in 2005. The NP dropped from 15,427 to 15,308 during 1986–2005, then rose to 15,765 blocks in 2010, and dropped again to 15,285 blocks in 2018. The increased NP indicated a relatively high degree of fragmentation of the landscapes. The data showed that the degree of fragmentation in the study area decreased from 1986 to 2005, rebounded in 2010, and was restored and optimized in 2018.

From 1986 to 2018, the PD in the study area was near one, and the overall changes were very small. The trend of PD from 1986 to 2010 was a slight decrease at first, an increase to 1.06, and then a decrease to 1.027 in 2018. This result demonstrated that the scale of the landscape pattern in the study area was basically the same as that of the natural landscape pattern at the unit of a hectare that can characterize the characteristics of the landscape pattern in the study area.

The LPI in the study area was approximately 8. The five-period data showed a slight downward trend from 1986 to 2018, with the highest in 1986 and the lowest in 2018. LPI is the ratio of the largest patch area to the entire landscape area. The extremely small LPI indicated that the flaky patches in the study area were small in area and generally fragmented. This finding was consistent with the terrain features dominated by mountains and hills in the study area. The slight-drop trend also indicated that human disturbances to ecosystems have been continuing. Disturbance to the largest patches needs paying particular attention to during ecological construction.

The PAFRAC remained at approximately 1.45 from 1986 to 2018, indicating that the patch shape in the study area was between a square and winding. From 1986 to 2010, the PAFRAC remained stable as a whole and decreased slightly from 2010 to 2018, showing that the shape of the patches had a tendency to evolve toward a square.

The SHDI and SHEI are types of landscape diversity characterizations. Table 3 shows that the SHDI was approximately 0.6 from 1986 to 2018, indicating that the landscape diversity and richness of the study area were high, and the landscape classes were rich. The SHEIs were all below 0.5, which reflected the low uniformity in the study area and the obvious dominant landscape. Both the SHDI and the SHEI remained stable from 1986 to 2010 and increased slightly from 2010 to 2018. This result demonstrated that the diversity and richness of the landscapes gradually increased in the 2010s.

From 1986 to 2018, the AI of the study area was greater than 87%, which demonstrated that the landscapes had a high aggregation degree. The AI decreased from 1986 to 2010 and increased to 87.79% from 2010 to 2018. This demonstrated that the connectivity between the various landscapes in the study area was enhanced, and the ecosystems could recover quickly after being disturbed.

## 4. Discussion

Ecosystems bring us food, natural fibers, a steady supply of clean water, regulation of pests and diseases, medicinal substances, recreation, and protection from natural hazards such as floods [44]. However, a substantially growing population, constantly expanding economic development, and massive requests for resources are altering the ecosystem in the planet [45], resulting in continuing environmental degradation. To keep the development of global sustainability, Stephen et al., introduced the planetary boundary (PB) concept in 2009 with the aim of defining the environmental limits within which humanity can safely operate, which has proved influential in sustainability policy [46]. The Chinese government is faced with the same problem to find the minimum space for wildlife to improve ecological functions and to ensure the sustainable supply of ecological goods and services [47]. In order to improve China’s ecological security and guide nature conservation in the future, the Chinese government initiated a new strategy, named ecological red lines (ERLs) [48]. The delimitations of ERLs were completed in 2021 in all the provinces and cities. The preliminarily delimited ERLs account for about 25% of the land area, covering the core ecological function areas, key areas of diversity distribution, ecologically sensitive areas, and fragile areas in the whole country [27], which answered for the first time how much ecological area needs to be strictly protected to achieve ecological civilization and sustainable development in China, providing a valuable policy instrument for improving nature conservation worldwide [48,49]. This concrete action in the construction of ecological civilization provides a case for the protection of ecological environment worldwide.

The studied ERL area in this paper is 14,875 km^2^, accounting for more than 1/3 of the total upper reaches of the Ganjiang River. The forests’ coverage rate reaches as high as 82%. The large patch area, high-proportion top successional biological types, decreased PAFRACs, and increased AIs of forests and grasslands all show that the study area is an ideal biodiversity conservation area and water conservation area. It also covers three national nature reserves, one provincial nature reserve, and 25 municipal or county-level nature reserves, which are especially protected with laws such as *Regulations of the People’s Republic of China on Nature Reserves.* Yet, we also found large areas of croplands, expanded human settlements, and patch fragmentation trends, which show obvious human disturbances to the ecosystem. Human disturbance often leads to negative impacts on wildlife in most cases. Hannah et al., mapped human disturbance in natural ecosystems and found that nearly three-quarters of the land on the planet was disturbed in some way in 1990s [50]. Doherty et al., reviewed 208 studies on 167 species from terrestrial and aquatic ecosystems and found that human disturbance had widespread impacts on the movements of birds, mammals, reptiles, amphibians, fish, and arthropods [51]. Steibl et al., observed that human disturbance negatively impacted the underlying food resource and habitat availability of hermit crabs on islands in their scientific reports [52]. Gaynor et al., reported in their research in *Science* that human disturbance had a strong effect on diurnal animals who increased their nocturnality by an average factor of 1.36 [53]. ERls are minimum ecological environmental spaces to move towards sustainable development [47]. Delimiting ERLs is only the starting point. How to strictly limit human activities in the disturbed ERLs should be the key to subsequent management.

However, this paper did not fully explain whether the delimited ERLs were sufficient for biodiversity protection and water conservation in the study area. Firstly, the data were from 30 m-resolution remote sensing images, which could only provide limited information about land use and land cover and were difficult to provide taxonomic information on species. Secondly, the methods were based on landscape ecology and the indexes selected were used to measure landscape features with scales above patches. Furthermore, the main topic of the paper is to evaluate the landscape changes. How these changes affected the ecological functions and services will be investigated in the following study.

## 5. Conclusions and Optimization Suggestion

Based on the needs to ensure the provision of ecosystem services, the characteristics and changes in the landscape ecological patterns in the ERL in the UR-GJ were assessed through satellite remote sensing data from 1986, 1995, 2005, 2010, and 2018. The conclusions are as follows:(1)The ecosystems in the ERL of the study area are effective in providing the ecosystem services of carbon sink function, biodiversity conservation, and water conservation function, especially due to the large percentage of forest cover. Forests are the largest and most important class of all the six ecological landscapes, accounting for more than 81% of the total study area.(2)The natural ecosystems in the study area keep relatively stable during 1986–2018. According to the indices at the landscape scales, the degree of landscape fragmentation in the study area decreased from 1986 to 2005. The diversity and richness of the landscape gradually increased from 2010 to 2018. The dominant characteristics of the forests were enhanced. The proportion of landscape types that have succeeded is very small. The ecosystems and aggregation can recover quickly after being disturbed by human activities.(3)According to the data from 2018, the study area was a good biological habitat area in the UR-GJ. The patch area was large. The dominant ecological types accounted for a high proportion, and most of them were top successional biological types. The dominant position of the forest landscapes was stable and increased, and there existed an obvious aggregation effect. The PAFRACs of all the six landscape types decreased, and the AIs of forests and grasslands increased. These findings demonstrated that during the period from 2010 to 2018, forest patches were gradually becoming connected, grasslands tended to be fragmented, and the patch shapes developed toward regularization. It was an ideal biodiversity conservation area and water conservation area.(4)There exist obvious human activity disturbances to the ecosystem in the ERL during 1986–2018. The proportion of croplands was approximately 11% of the total area. The analysis of ecological succession found that human settlements expanded to ecological areas from 2010 to 2018. The ecological index of the patch scales also showed a fragmentation trend from 1995 to 2010, and the phenomenon of the croplands and settlement landscapes invading the natural ecosystem. The establishment of effectively designed and managed ERLs will have a positive effect in maintaining the natural ecosystems that have been providing substantial ecosystem services to the study area.

Based on the findings of our study, we provide the following suggestions for designing and implementing effective management of the ERLs:(1)The conversion of large-scale croplands to forests could be the key to maintaining the stability of the ecosystem in the future in the ERL. Croplands account for approximately 11% of the total area. It is important to return croplands to forests so as to reduce the impact of human activities on the ecosystem. Croplands are easily driven by policy and the economy, and the proportion of croplands in the study area is more scattered, less connected, and low in aggregation. The patch of croplands is relatively fragmented and has difficulty forming a scale advantage. The natural conditions have an obvious restrictive effect on agricultural development. Expanding human settlements indicate that a combination of increasing yields from highly productive agricultural land and converting marginal grasslands and agricultural land to forests would yield greater benefits from the delivery of essential ecosystem services such as watershed protection and biodiversity conservation. Under the expansion condition of human settlements, the transformation of large-scale croplands to forests could be the key to maintaining the stability of the ecosystem in the study area.(2)The ERLs need to be protected against ecological disruption, with measures such as strictly prohibiting human production activities in the mountains to facilitate afforestation. This will allow the disturbed landscapes to recover under the natural drives of good climate, soils, and geology and gradually reach successional climaxes. These measures will increase the proportion of natural forests and continuously optimize the ecosystem functions of the study area.(3)According to the characteristics of ecological patches in the study area, it is recommended to pay special attention to the establishment and maintenance of the largest patch ecosystems during ecological management. The LPI in the study area was only approximately 8, and it declined in 2005. Under fragmentation conditions, the establishment of the largest patch should be paid attention to so as to optimize management, and this can help realize contiguous development and the aggregation effect of key landscape types such as forests and grasslands.

## Figures and Tables

**Figure 1 ijerph-19-11530-f001:**
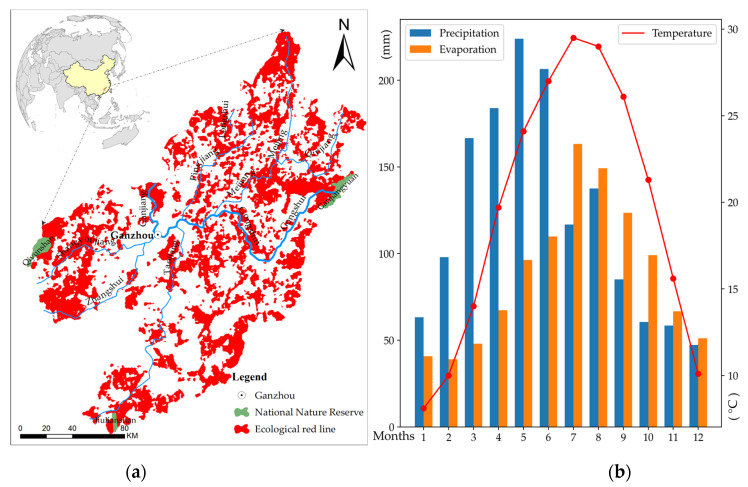
Study area. (**a**) Location of ecological red lines (ERLs) in the upper reaches of the Ganjiang River (UR-GJ, from the Jiangxi Provincial Government). (**b**) Monthly precipitation (1951–2020), monthly evaporation (1962–2019), and monthly average temperature (1951–2019) at Ganzhou meteorological station (from http://data.cma.cn accessed on 10 January 2020).

**Figure 2 ijerph-19-11530-f002:**
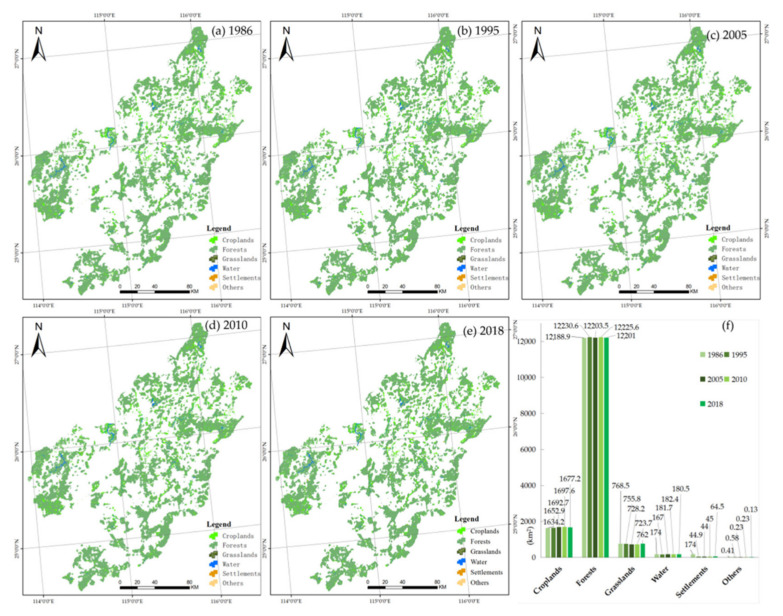
Landscape ecological-type maps of the ERL in the UR-GJ. (**a**–**e**) are the maps of landscape ecological types in 1986, 1995, 2005, 2010, and 2018, respectively; (**f**) is the area histogram of landscape ecological types in 1986–2018.

**Figure 3 ijerph-19-11530-f003:**
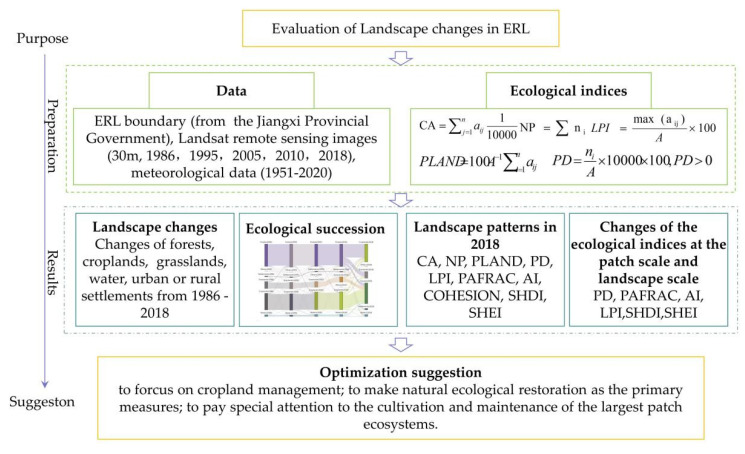
The framework for landscape changes and optimization in ERLS in the UR-GJ.

**Figure 4 ijerph-19-11530-f004:**
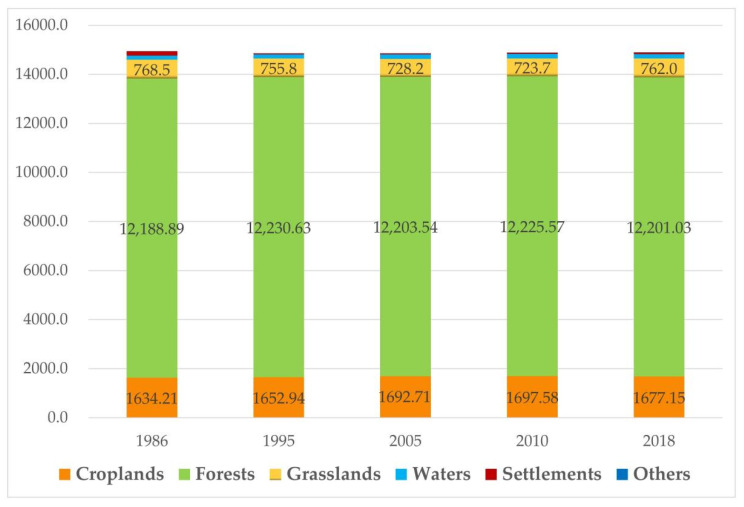
Statistics of the landscape changes from 1986 to 2018.

**Figure 5 ijerph-19-11530-f005:**
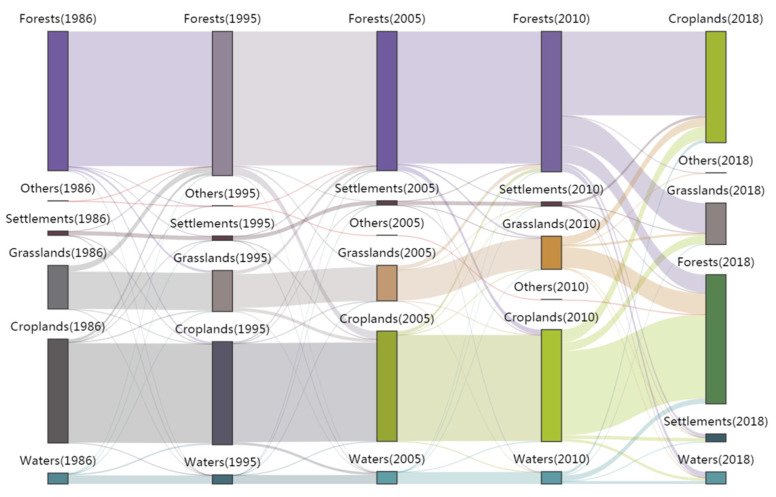
Sankey map of the ecological succession in the study area.

**Figure 6 ijerph-19-11530-f006:**
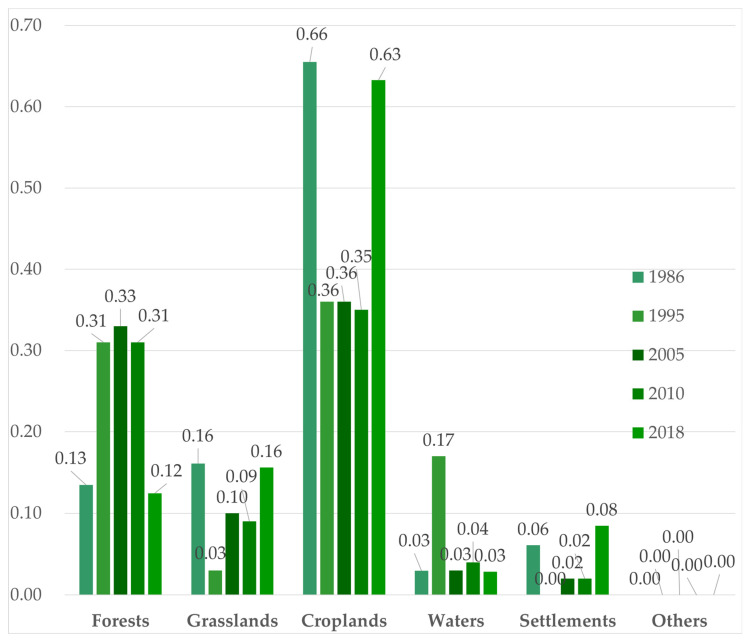
Changes in the PD values from 1986 to 2018.

**Figure 7 ijerph-19-11530-f007:**
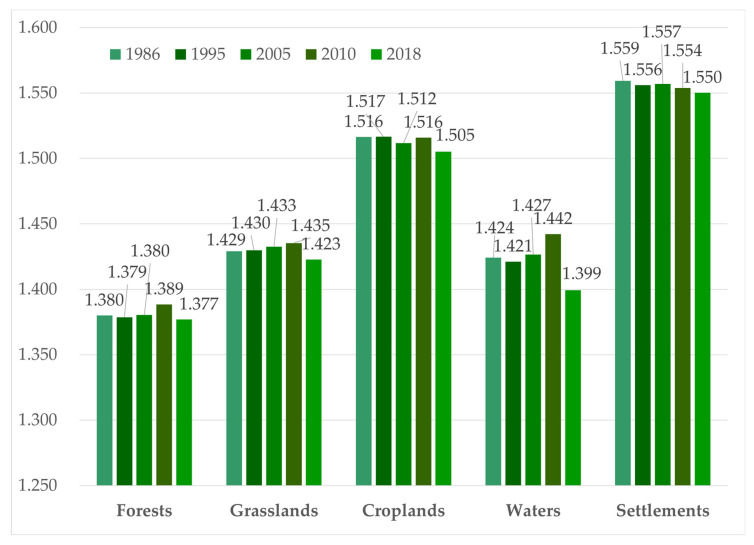
Changes in the PAFRAC from 1986 to 2018.

**Figure 8 ijerph-19-11530-f008:**
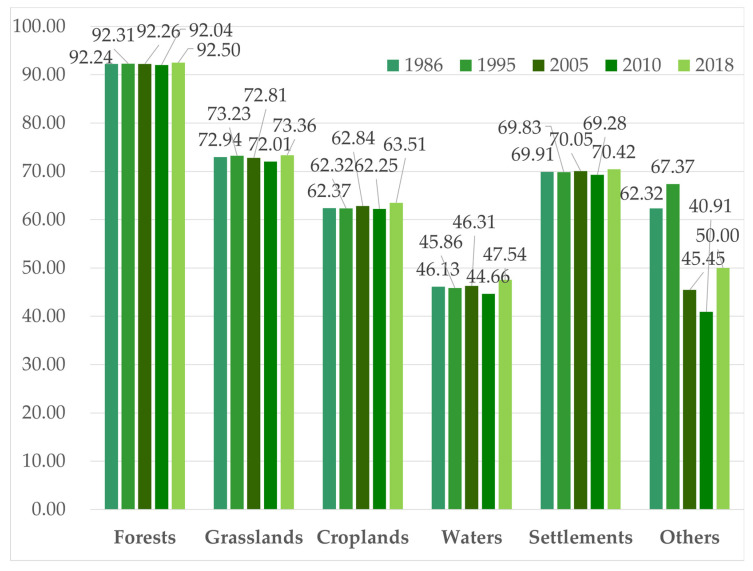
Changes in the AI from 1986 to 2018.

**Table 1 ijerph-19-11530-t001:** Case study of delimiting ERLs in China.

Authors	Approaches	Places	References
Wang ad Nan	Indicator: water conservation, soil and water conservation, biodiversity protection, water resource protection, and flood regulation as indicators. Method: GIS	Anhui Province	[33]
Yan et al.	Methods: evaluation of the importance of ecosystem services	Jiangsu Province	[34]
Yang et al.	Method: the least-resistance model for maintaining the ecological security pattern	Jiangxi Province	[35]
Xie et al.	Method: GIS and ecological sensitivity evaluation	Guizhou Province	[36]
Xu et al.	Method: a comprehensive multifactor analysis	Macao	[37]

**Table 2 ijerph-19-11530-t002:** Area and proportion of each landscape type from 1986 to 2018.

Year	1986		1995		2005		2010		2018	
Landscape Types	Area (km^2^)	Proportion (%)	Area (km^2^)	Proportion (%)	Area (km^2^)	Proportion (%)	Area (km^2^)	Proportion (%)	Area (km^2^)	Proportion (%)
Croplands	1634.2	10.9	1652.9	11.1	1692.7	11.4	1697.6	11.4	1677.2	11.3
Forests	12,188.9	81.6	12,230.6	82.4	12,203.5	82.2	12,225.6	82.2	12,201.0	82.0
Grasslands	768.5	5.1	755.8	5.1	728.2	4.9	723.7	4.9	762.0	5.1
Water	174.0	1.2	167.0	1.1	181.7	1.2	182.4	1.2	180.5	1.2
Settlements	174.0	1.2	44.9	0.3	44.0	0.3	45.0	0.3	64.5	0.4
Others	0.41	0.003	0.580	0.000	0.230	0.000	0.230	0.000	0.130	0.001

**Table 3 ijerph-19-11530-t003:** Index statistics of the patch scales in 2018.

Landscapes	CA (hm^2^)	PLAND (%)	NP	PD	LPI	PAFRAC	COHESION	AI
Croplands	167,715	11.3	9420	0.6	0.1	1.51	87.8	63.5
Forests	1,220,103	82.0	1853	0.1	7.9	1.38	99.5	92.5
Grasslands	76,200	5.1	2325	0.2	0.2	1.42	90.3	73.4
Water	18,052	1.2	423	0.0	0.2	1.55	94.4	70.4
Settlement	6445	0.4	1260	0.1	0.0	1.40	62.9	47.5
Others	13	0.0	4	0.0	0.0	N/A	46.1	50.0

**Table 4 ijerph-19-11530-t004:** Ecological indices at the landscape scale from 1986 to 2018.

Years	NP	PD	LPI	PAFRAC	SHDI	SHEI	AI
1986	15,427	1.042	8.060	1.452	0.628	0.350	87.535
1995	15,338	1.031	8.045	1.452	0.623	0.348	87.632
2005	15,308	1.029	7.958	1.452	0.627	0.350	87.569
2010	15,765	1.060	7.949	1.459	0.626	0.349	87.273
2018	15,285	1.027	7.853	1.442	0.638	0.356	87.789

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
