# Peer review of "Landscape Changes and Optimization in an Ecological Red Line Area: A Case Study in the Upper Reaches of the Ganjiang River"

_ijerph, 2022, doi:10.3390/ijerph191811530_

Round 1

Reviewer 1 Report

The article is relevant and the topic is worthy of investigation.  Likewise, the methodology is rigorous and has scientific soundness.  However, there are a couple of issues that should be improved to strengthen the level of academic and scientific scholarship of the manuscript.  Therefore, I recommend a moderate revision of the text.  If both questions are correctly incorporated by the authors, I believe that the article could be considered for publication in the journal.  Below I detail the issues that should be addressed by the authors.

SCIENTIFIC DISCUSSION 

A scientific discussion section after the presentation of results is missing in the paper.  The authors must include a section between the results and conclusions sections in which the results obtained are discussed in comparison with previous studies, indicating to what extent the results obtained in the investigation corroborate, contradict or improve the results of said previous studies.  In this context, it is also possible to incorporate a more international view of the problem and the possible application of the results obtained beyond the specific area of ​​study, since the approach presented for research is excessively local.  Finally, this new section should also address, from a more self-critical point of view, the limitations of the work carried out that can be improved in future lines of research.

REVIEW OF THE STATE OF THE ART

The review of the state of the art of the introduction is a bit scarce and excessively local from the methodological point of view (the authors introduce the subject by making a brief literary review in a single paragraph to then directly address the specific issue of their case study in China).  This somewhat impoverishes the level of scientific and academic scholarship of the manuscript.  The authors should cite a greater number of non-Chinese studying using alternative methodological approaches for the analysis of ecological problems based on landscape GIS indicators (there are many approaches beyond those cited, e.g.  https://doi.org/10.3390/su12187589,  https://doi.org/10.3390/su10061820 or https://doi.org/10.3390/rs11243041 ) and then justify why their proposal improves the currently existing methodologies or why it contributes to filling a gap that currently exists in this field of research.

Reviewer 2 Report

See my detailed editorial suggestions submitted separately.

Round 2

Reviewer 2 Report

None.  already sent earlier comments.